# Unsupervised Image Registration towards Enhancing Performance and Explainability in Cardiac and Brain Image Analysis

**DOI:** 10.3390/s22062125

**Published:** 2022-03-09

**Authors:** Chengjia Wang, Guang Yang, Giorgos Papanastasiou

**Affiliations:** 1Edinburgh Imaging Facility QMRI, Centre for Cardiovascular Science, University of Edinburgh, Edinburgh EH16 4TJ, UK; chengjia.wang@hw.ac.uk; 2Faculty of Medicine, National Heart & Lung Institute, Imperial College London, London SW7 2BX, UK; 3School of Computer Science and Electronic Engineering, University of Essex, Colchester CO4 3SQ, UK

**Keywords:** multi-modality image registration, unsupervised image registration, deep learning, inverse-consistency, explainable deep learning

## Abstract

Magnetic Resonance Imaging (MRI) typically recruits multiple sequences (defined here as “modalities”). As each modality is designed to offer different anatomical and functional clinical information, there are evident disparities in the imaging content across modalities. Inter- and intra-modality affine and non-rigid image registration is an essential medical image analysis process in clinical imaging, as for example before imaging biomarkers need to be derived and clinically evaluated across different MRI modalities, time phases and slices. Although commonly needed in real clinical scenarios, affine and non-rigid image registration is not extensively investigated using a single unsupervised model architecture. In our work, we present an unsupervised deep learning registration methodology that can accurately model affine and non-rigid transformations, simultaneously. Moreover, inverse-consistency is a fundamental inter-modality registration property that is not considered in deep learning registration algorithms. To address inverse consistency, our methodology performs bi-directional cross-modality image synthesis to learn modality-invariant latent representations, and involves two factorised transformation networks (one per each encoder-decoder channel) and an inverse-consistency loss to learn topology-preserving anatomical transformations. Overall, our model (named “FIRE”) shows improved performances against the reference standard baseline method (i.e., Symmetric Normalization implemented using the ANTs toolbox) on multi-modality brain 2D and 3D MRI and intra-modality cardiac 4D MRI data experiments. We focus on explaining model-data components to enhance model explainability in medical image registration. On computational time experiments, we show that the FIRE model performs on a memory-saving mode, as it can inherently learn topology-preserving image registration directly in the training phase. We therefore demonstrate an efficient and versatile registration technique that can have merit in multi-modal image registrations in the clinical setting.

## 1. Introduction

Clinical decision-making from magnetic resonance imaging (MRI) is based on combining anatomical and functional information across multiple MRI sequences (defined throughout as “modalities”). Multiple imaging biomarkers can be derived across different MR modalities and organ areas. This makes image registration an important MR image analysis process, as it is commonly required to “pair” images from different modalities, time points and slices. Hence, both intra- and inter-modality image registration are essential components in clinical MR image analysis [1], and finds wide use in longitudinal analysis and multi-modal image fusion [2].

Although numerous deep learning (DL) methods have been devised for medical image analysis [2], DL-based image registration tasks have been relatively less explored [3,4]. Among DL-based registration studies, supervised learning-based methods showed promising results. The main disadvantage of supervised learning is that it necessitates laborious and time-consuming manual annotations for model training, even as it is difficult to design generalised frameworks [5,6,7]. Unsupervised DL-based image registration methods gain increasing popularity, as they aim to overcome the need of training datasets with ground truth annotations [3,4]. However, unsupervised learning has been mainly investigated on intra(single)-modal image registration [3,4,8] and on either 3D volumes [9,10,11,12,13] or 2D images [14,15,16,17,18]. Also, previous unsupervised learning methods involve affine registration before training, which is a laborious, time-consuming and computationally expensive task [2,3,4,9,10,11,12,13,14,15,16,17,18].

To date, most previous unsupervised DL methods can perform either affine or non-rigid registration [3,4,9,10,12,13,14,15,16,17,18]. To our knowledge, the only unsupervised DL study that has developed a method for both affine and non-rigid registrations is by de Vos et al. [11]. In this work, DL modelling was performed using two autonomous models in the analysis pipeline to address both affine and non-rigid registrations, whilst also requiring affine transformations before training. Inverse-consistency-based transformation models show improved ability to preserve the contextual and topology information in image registration [19]. Note that a deformable image registration between two images is called inverse consistent, when the correspondence between images is invariant to the order of the choice of the source and target image [19,20,21].

In our work, we demonstrate a bi-directional unsupervised DL model that is capable of performing multi-modal (n-D, where n = 2–4) affine and non-rigid image transformations. As will be discussed in the next sections, we model n-D affine and non-rigid registrations through bi-directional cross-modality synthesis and inverse-consistent spatial transformations. Unlike previous unsupervised learning studies that focus on estimating asymmetric transformations and cannot preserve topology [2,3,4,9,10,11,12,13,14,15,16,17,18], our proposed technique is efficient for both affine and non-rigid image registrations, as demonstrated in multi-modal brain and cardiac image registration experiments.

### 1.1. Motivation

Unsupervised DL techniques have considerable potential in medical image registration. In this paper we combine up-to-date progress in unsupervised learning with strong prior knowledge about medical image registration: reaching high registration performance, we synchronically perform multi-modal image synthesis and factorised spatial transformations (one per each encoder-decoder channel). This operation allows us to efficiently reach inverse-consistent multi-modal affine and non-rigid image registration of multi-dimensional medical imaging data, making our work efficient and versatile for the medical imaging community.

The FIRE model provides a generalised architecture for registration applications, as the synthesis components of the FIRE model can be customised and re-trained if/when data from additional medical imaging modalities would need to be co-registered. The synthesis factor captures global image modality information, determining how organ topology is rendered in the target image. Between other things, maintaining a representation of the modality characteristics in the synthesis factor of the model provides the ability to potentially model data from multiple modalities. We evaluate this function using the multi-modal MR data of variable anatomical information (see also the “model explainability” section).

Finally, the incorporation of factorised spatial transformations in the FIRE model allows us to learn varied image transformations. We examine whether the FIRE architecture and its associated loss functions can directly output transformation fields for both affine and non-rigid registrations. We demonstrate that the FIRE model architecture efficiently learns to produce inverse-consistent deformations that are intrinsically topology-preserving.

### 1.2. Overview of the Proposed Approach

Learning image representations and spatial transformations through image synthesis is an area of recent work in medical image analysis [20]. However, there was no previous consideration about the precision of the synthesis process and the versatility of the spatial transformations. This is important in medical image analysis, as both affine and deformable registrations are commonly required for a typical clinical dataset.

Following our previous work [22], we demonstrate a versatile registration method called “FIRE” by explicitly incorporating spatial transformation into our cross-domain bi-directional image synthesis (Figure 1). We show that our method can robustly model 2D/3D and 4D registrations when examined on multi- and intra-modality brain and cardiac MRI, respectively. We provide thorough explanations regarding our model explainability in medical image registration. Moreover, we demonstrate the efficiency of our method on computational time experiments.

### 1.3. Contributions

Our main contributions are as follows:With the use of two spatial transformation networks factorised into our cross-domain synthesis, the “火”-shape FIRE model architecture can be trained (through our combined loss functions) end to end, to perform inverse-consistent unsupervised registrations using a single DL network for both inter- and intra-modality registration.Incorporating a new affine regularisation loss to eliminate non-rigid transformation where needed, we show that our FIRE model simultaneously learns both affine and non-rigid transformations.Overall, the novel FIRE model loss functions and training procedure are designed to allow simultaneous learning for synthesis and registration of 2D, 3D and 4D multi-modal MR data.We show that our FIRE model is explainable and provides a comprehensive (affine and non-rigid) framework for future improvements and generalisation.

## 2. Related Work

In this section we review previous work on registration methods using DL (Section 2.1). We then review inverse-consistent DL transformations (Section 2.2). Finally, we review methods that solve inter-modality registration through image synthesis (Section 2.3).

Since we detail the background and motivation of our work against previous studies in the Introduction section of this paper, here we focus on discussing similarities and differences of our FIRE model against previous methods that are relevant to our application domain.

### 2.1. Registration Methods Using DL

Early DL registration methods were mostly adaptations of the conventional image alignment scheme, where DL was used to extract convolutional features for guiding correspondence detection across subjects [23], or to learn new similarity metrics across images [24,25].

Reinforcement learning methods have shown promising results, but they are still designed to solve image registration through iterative processes, making them computationally expensive and time-consuming [26,27,28]. As discussed, although DL-supervised learning-based methods have also been promising for image registration, one of their main disadvantages is that they require laborious and time-consuming manual annotations [5,6,7,29].

To overcome the limitations of supervised learning methods, unsupervised learning approaches were recently developed that learn image registration by focusing to minimise the loss between the deformed image and fixed (target) image. The development of a spatial transformer network (STN) in 2015 by Jaderberg et al. has inspired many unsupervised learning methods as it can be inserted into existing DL models [8]. The STN allows explicit manipulation of images within a network and can potentially perform image similarity loss calculations during the training and testing process. Balakrishnan et al. demonstrated Voxelmorph, a 3D medical image registration algorithm using STN within a convolutional neural network (CNN), in which parameters are learned by the local cross-correlation loss function [30,31]. Krebs et al. proposed a probabilistic formulation of the registration problem through unsupervised learning, in which an STN was used within a conditional variational autoencoder [32]. Kim et al. developed CycleMorph, a cycle-consistent DL method for deformable image registration (through an STN component) and demonstrated accurate registration results [33]. Unlike previous techniques [30,31,32,34,35], the CycleMorph model incorporated inverse consistency directly in the training phase [33] to improve model efficiency and robustness.

To date, the only unsupervised DL study that has developed a method for both affine and non-rigid registrations is by de Vos et al. [11]. However, in this work, DL modelling required the incorporation of two autonomous models in the analysis to address both affine and non-rigid registrations, whilst also requiring affine transformations as part of the modelling process (therefore adding complexity and computational cost).

### 2.2. Inverse-Consistent DL Transformations

Most recent studies are mainly focusing on performing non-rigid deformations [30,31,32,33,34,35,36], whilst requiring computationally expensive and laborious affine transformations in pre-processing using ANTs [30,36,37] or some other software application [20]. Note that affine transformation is inherently inverse-consistent because of the invertibility of the transformation matrix. In the past decade, inverse consistency has mainly been achieved by incorporating diffeomorphic deformation fields [37]. Some of the most popular examples include the diffeomorphic Demons algorithms [38,39].

Several DL diffeomorphic architectures have recently been proposed, such as the Voxelmorph [30,31,34,35], the Quicksilver [38] and the CycleMorph [33]. However, most of these unsupervised diffeomorphic models were tested on intra-modality brain data [30,31,34,35,38]. As already discussed in Section 2.1, the CycleMorph model is still considered the only technique to effectively incorporate inverse consistency directly in the training phase, whilst it was tested on multi-modal (brain MRI and liver CT data) medical imaging data. In our work, we define the output deformation fields with simple inverse-consistency constraints, devising a model that can learn topology-preserving registratrion in an expandable and versatile architecture. It may be important to emphasise that Nielsen et al. recently concluded that the diffeomorphism fails to account for local discontinuities in brain images with pathological conditions [39]. Despite this finding, we should highlight that it is straightforward to integrate diffeomorphic layer components into the FIRE framework if deemed as appropriate (for example, the Voxelmorph velocity field [30,31], or the CycleMorph homeomorphic mapping [33] could potentially be incorporated).

In clinical practice, image registration tasks commonly involve comprehensive image deformations, i.e., a combination of affine and non-rigid registration. In our study, we demonstrate a versatile and efficient DL model that can perform both affine and non-rigid (deformable) registration, and which can learn to produce inverse-consistent and intrinsically topology-preserving image deformations.

### 2.3. Image Synthesis DL Methods

Image synthesis-based methods have potential in recent DL-based inter-modality registration algorithms. Alongside the recent developments on adversarial cycle-consistent loss [40], most recent methods focus on using CycleGAN for synthesis-based inter-modality registration [20,41]. Wei et al. performed image registration by devising CycleGAN-based image synthesis as a separate processing step prior to registration [41]. Qin et al. proposed a bi-directional unsupervised training framework, which is conceptually close to our model. However, our FIRE model is simultaneously learning synthesis and registration on a memory-saving architecture that requires less hyperparameters when balancing all different losses (see Section 3). Moreover, the FIRE model is evaluated on 2D, 3D and 4D data involving different organs (brain and heart; note that 4D here denotes periodic heart deformations observed across different time phases of cardiac cine-MRI).

## 3. Methods

Our proposed model can be described as a bi-directional cross-domain image synthesis structure, with two factorised spatial transformation components (one per each encoder-decoder channel) [22].

Each time two images x^A^ and x^B^ are processed, the FIRE model learns φ^A→B^ and φ^B→A^ transformations, warping x^A^ and x^B^ into x^A^ ◦ φ^A→B^ and x^B^ ◦ φ^B→A^, via the transformation networks T^A→B^ and T^B→A^. In parallel, two synthesis encoders G(x^A^) and G(x^B^) extract modality-invariant latent representations through the synthesis decoders F^A→B^ and F^B→A^, mapping the representations extracted by G(x^A^) and G(x^B^) to the synthesed images $\hat{x}$^B^ and $\hat{x}$^A^, respectively.

### 3.1. Architecture

Our model involves the following components (Figure 1): (a) one synthesis encoder G, which extracts modality-invariant representations G(x^A^) and G(x^B^); (b) two synthesis decoders F^A→B^ and F^B→A^, which map G(x^A^) and G(x^B^) to $\hat{x}$^B^ = F^A→B^(G(x^A^)) and $\hat{x}$^A^ = F^B→A^(G(x^B^)) (synthesised images); and (c) two transformation networks, T^A→B^ and T^B→A^, which model the transformation fields φ^A→B^ = T^A→B^(G(x^A^), G(x^B^)) and φ^B→A^ = T^B→A^(G(x^B^), G(x^A^)). Note that during training, G(x^A^) and G(x^B^) are warped into G(x^A^) ◦ φ^A→B^ and G(x^B^) ◦ φ^B→A^ before being used to generate the synthesised images: ${\hat{x}}_{T}^{B}$ = F^A→B^(G(x^A^) ◦ φ^A→B^) and ${\hat{x}}_{T}^{A}$ = F^B→A^(G(x^B^) ◦ φ^B→A^).

### 3.2. Encoder and Decoder for Synthesis

The structures of the encoder and decoders were inspired by the design of the CycleGAN [41], which includes a set of downsampling and upsampling convolutional layers connected through a series of Resnet blocks.

We modified the architecture of the CycleGAN generator by adding two decoders in the generator and by eliminating the discriminator (instead of having one decoder in the generator followed by a discriminator, as in the original CycleGAN, Figure 1 and Figure 2) [41]. In our model, we did not need to recruit adversarial learning through the discriminator, since by involving two decoders and geometrical mapping via our STN-derived deformations in the cross-domain synthesis learning, our model was designed to match the (pixel intensity and geometrical) distribution of generated images to the data distribution in either target domain. 

The encoder G contained one convolutional block (7 × 7, 64) with stride 1, two downsampling convolutional layers (3 × 3, 128; 3 × 3, 256) with stride 2 and four Resnet blocks of the same size (3 × 3, 256) with stride 1. The decoders contained four consecutive Resnet blocks of the same size (3 × 3, 256) with stride 1, two convolutional layers used for upsampling (3 × 3, 128; 3 × 3, 64) with stride 1 and a convolutional block (7 × 7, 3) with stride 1. The synthesised output was finally produced by a 1 × 1 convolutional layer, followed by a Tanh layer [22]. 

The downsampling layers (in the encoder) encode the images down to an abstract latent representation. In this process, the model learns weights that enable the reducing of the spatial resolution of the feature (image pixel) maps. The output feature maps derived from downsampling are then passed through a series of Resnet blocks with skip connections to further interpret and process their underlying information while addressing the vanishing gradient problem that may occur in large networks [20,41]. Through the upsampling layers, the model learns to transpose (reverse) convolutions, decoding the latent representations back to the size of the ouput image. 

### 3.3. Transformation Network

The transformation networks of our model, T^A→B^ and T^B→A^, can learn both affine and non-rigid transformations (denoted as φ_af_ and φ_nr_, respectively, Figure 3). Each transformation network consists of a subnetwork for affine transformation followed by a subnetwork for nonrigid transformation, as shown in Figure 3.

To derive deeper geometrical inter-relationships from the processed data, we modified the start of the STN structure by adding three convolutional layers with instance normalization and standard LeakyReLU layers (Figure 3). In specific, the affine transformation network T_af_ uses an initial convolutional layer of size (1 × 8, 7) and stride 2, followed by two “convolutional layer + ReLU + Instance Norm” blocks (the size of the first and second convolutional layers used were 8 × 10, 5 and 16 × 20, 2, respectively, with stride 2). Following the two “convolutional layer + ReLU + Instance Norm” blocks, two Dense (fully connected) layers were recruited to compute the parameters of the affine transformation matrix (2 × 3 for 2D; 3 × 4 for 3D transformations) [8]. Essentially, the input of the affine subnetwork is the concatenated features obtained from the two input images. The convolutional features derived from the third convolutional layer were resampled into a fixed-size feature vector using Global Average Pooling [8,22], followed by two fully connected layers that were used to generate the transformation matrix.

Non-rigid transformation network T_nr_^A→B^ receives the affinely transformed feature G(x^A^) ◦ T_nr_^A→B^ and the untransformed feature G(x^B^) as inputs, and subsequently processes them in parallel layers (Figure 3). The features extracted are finally concatenated and used as inputs into a ResNet block with instance normalization and LeakyReLU layers. To produce the non-rigid deformation φ_nr_^A→B^, a final convolutional layer and a Tanh activation function are implemented. The Tanh layer is performed on a normalised coordinate system: a coordinate *p* ∈ [−1, 1] n exists for an n-D image [22].

As described, we factorised two separate transformation networks for each encode-decoder channel in our FIRE model, which means that the above processes described for the A→B (affine and non-rigid) transformations are identical for the B→A affine and non-rigid transformations as well.

### 3.4. Training Procedure and Overall Loss Function

Our FIRE model is trained for simultaneously learning mutually inverse registration and synthesis tasks across both “A→B” and “B→A” directions.

The A→B image synthesis process synthesised images ${\hat{x}}^{B}$ and ${\hat{x}}_{T}^{B}$, so that ${\hat{x}}^{B}$ is registered with x^A^ and ${\hat{x}}_{T}^{B}$ is identical to the target image x^B^. The A→B transformation learned by T^A→B^ is applied to the features G(x^A^) for synthesis purposes, and subsequently to image x^A^ for registration.

The B→A image synthesis process performs backward registration and synthesis. All parameters of the entire network are updated end to end in the training process, and only G and a single transformation network are required to deform the moving image in the testing phase, for either modality input image.

To train the proposed FIRE model, a synthesis loss (ℒsyn and a registration loss (ℒreg) are recruited. A regularisation process (ℛ) is introduced to perform topology-preserving deformation and spatial smoothing [22]. Combining these terms, the overall loss function of the proposed model is:(1)ℒ=ℒsyn+ℒreg+ℛ

As described in the next subsections, we formulate most of the terms in ℒsyn, ℒreg and ℛ using root-mean-square (RMS) calculated on normalised inputs. As a result, apart from regularising the smoothness of the non-rigid deformation field, no hyperparameter is required to balance different losses, as detailed below. 

### 3.5. Synthesis Loss

There are four synthesis loss terms involved, each one supporting a different purpose (see Figure 4). First, to perform accurate cross-domain synthesis, the following synthesis accuracy loss was defined (by implementing the RMS error):(2)$$ℒ\mathrm{syn},\mathrm{acc}=\mathrm{RMS}({\hat{x}}_{T}^{B},\;x^{B})+\mathrm{RMS}({\hat{x}}_{T}^{A},\;x^{A})$$

Note that for image synthesis, the synthesised images ${\hat{x}}_{T}^{A}$ and ${\hat{x}}_{T}^{B}$ aim to be identical to the target images x^A^ and x^B^, respectively, which is defined in Equation (2).

Second, G is designed to learn modality-invariant features. To represent this, we defined the following feature loss, which aims to minimise the RMS error between G(x^A^) and G(x^B^) $◦\;\varphi^{B\to A}$, as well as G(x^B^) and G(x^A^) $◦\;\varphi^{A\to B}$:(3)$$ℒ\mathrm{syn},\mathrm{fea}=\mathrm{RMS}(G(x^{A}),\;G(x^{B})◦\varphi^{B\to A})+\mathrm{RMS}(G(x^{B}),\;G(x^{A})◦\varphi^{A\to B})$$

Equation (3) was designed to minimise the error between the G outcomes from either modality and the transformations learned in the G outcomes of the other modality. 

Third, the cycle-consistency loss used in CycleGAN [20,41] has been proven to be critical for its superior performance in cross-domain image synthesis. For robust cross-modality synthesis performance, a cycle-consistency loss was therefore designed:(4)$$ℒ\mathrm{syn},\mathrm{cyc}=\mathrm{RMS}(F^{B\to A}\;(G({\hat{x}}^{B})),\;x^{A})+\mathrm{RMS}(F^{A\to B}\;(G({\hat{x}}^{A})),\;x^{B})$$

Essentially, Equation (4) encourages F^B→A^ (G(${\hat{x}}^{B}$)) = x^A^ and F^A→B^ (G(${\hat{x}}^{A}$)) = x^B^, enforcing these mappings to be reverses of each other (by minimising the error of either mapping procedure to the inputs x^A^ and x^B^). 

Finally, aligning x and $\hat{x}$ is important to transfer the geometric correspondence from either x^A^ or x^B^ (learned through the transformation networks), and to the synthesis process. To this end, the synthesis alignment loss was defined as: (5)$$ℒ\mathrm{syn},\mathrm{align}=\mathrm{RMS}(G(x^{A}),\;G({\hat{x}}^{B}))+\mathrm{RMS}(G(x^{B}),\;G({\hat{x}}^{A}))$$

The entire FIRE synthesis loss was [22] (Figure 4):(6)ℒsyn=ℒsyn,acc+ℒsyn,fea+ℒsyn,cyc+ℒsyn,align

### 3.6. Registration Loss

To perform synthesis and registration, features extracted by G were transformed and registration was then performed through the following transformations to the input images x^A^ and x^B^: φ^A→B^ = φ_af_^A→B^ ◦ φ_nr_^A→B^ and φ^B→A^ = φ_af_^B→A^ ◦ φ_nr_^B→A^. Because the output of the synthesis process F^•^^→^^•^ (G(·)) is aligned with its input, the synthesised image obtained from a transformed x^A^ image should be identical to x^B^. Based on this, we defined the following registration accuracy loss:(7)$$ℒ\mathrm{reg},\mathrm{acc}=\mathrm{RMS}(F^{A\to B}(G(x^{A})◦\varphi^{A\to B}),\;x^{B})+\mathrm{RMS}(F^{B\to A}\;(G(x^{B})◦\varphi^{B\to A}),\;x^{A})$$

Finally, for inverse-consistent registration, the transformations φ^A→B^ and φ^B→A^ should be mutually inverse; thus, the following inverse-consistency loss was defined:(8)$$ℒ\mathrm{reg},\mathrm{ic}=\mathrm{RMS}(x^{A},\;x^{A}◦\varphi^{A\to B}◦\varphi^{B\to A})+\mathrm{RMS}(x^{B},\;x^{B}◦\varphi^{B\to A}◦\varphi^{A\to B})$$

This inverse-consistent loss can be seen as a transformation-oriented cycle-consistency loss, which encourages the composition of mutual mappings from the moving image to the fixed image on a bi-directional mode. For example, for x^A^ (moving image) to x^B^ (fixed image) registration, it minimises the error for both $\varphi^{A\to B}$ and $\varphi^{B\to A}$ mappings. 

The overall entire registration loss was computed as:(9)ℒreg=ℒreg,acc+ℒreg,ic

In practice, the non-rigid transformation field φ_nr_ was calculated on G(x) and linearly resampled before being applied to image x.

### 3.7. Regularisation

Inspired by conventional image registration, in which regularisation is commonly an integral part [11,22], we also add regularisation terms in both affine and non-rigid transformations. The main novelty in terms of regularisation in our work is that we incorporate separate regularization terms for both affine and non-rigid transformations in our model. At that stage, it is important to note a fundamental difference between affine and non-rigid transformation [11,30,31,32,33]: affine registration recruits global transformation fields, whilst non-rigid registration recruits mainly local transformation fields [11]. Non-rigid registration may contain affine transformations. However, affine registration should not involve non-rigid transformations.

In that context, we first add a bending energy penalty term to reinforce smooth displacements for both non-rigid transformation fields φ_nr_^A→B^ and φ_nr_^B→A^, as follows:(10)$$ℛ\mathrm{smooth}={∥\nabla^{2}\;{\varphi_{\mathrm{nr}}}^{A\to B}∥}^{2}+{∥\nabla^{2}\;{\varphi_{\mathrm{nr}}}^{B\to A}∥}^{2}$$
where ∇ is the Laplacian operator.

Second, because our FIRE model is designed to also calculate affine registrations, we propose to induce a separate regularization term to eliminate non-rigid transformations when affine transformation is present in the data. During image synthesis, the affinely transformed features G(x^A^) ◦ φ_af_^A→B^ and G(x^B^) ◦ φ_af_^B→A^ can be used as inputs into the synthesis decoders to obtain F^A→B^(G(x^A^) ◦ φ_af_ ^A→B^) and F^B→A^(G(x^B^) ◦ φ_af_^B→A^). The regularisation of the synthesis is then computed as:(11)$$ℛ\mathrm{syn}=\mathrm{RMS}(x^{B},\;F^{A\to B}(G(x^{A})◦{\varphi_{\mathrm{af}}}^{A\to B}))+\mathrm{RMS}(x^{A},\;F^{B\to A}(G(x^{B})◦{\varphi_{\mathrm{af}}}^{B\to A}))$$

Similarly, a regularisation for registration, ℛreg, is defined using the affinely deformed x^A^ and x^B^:(12)$$ℛ\mathrm{reg}=\mathrm{RMS}(x^{B},\;F^{A\to B}(G(x^{A}◦{\varphi_{\mathrm{af}}}^{A\to B})))+\mathrm{RMS}(x^{A},\;F^{B\to A}(G(x^{B}◦{\varphi_{\mathrm{af}}}^{B\to A})))$$

The overall regularisation of the FIRE model was [22]:(13)ℛ=ℛsyn+ℛreg+λℛsmooth
where λ is the scaling parameter for ℛsmooth, which is the only hyperparameter used in the FIRE loss. We should note that to register n-D images, λ = 2^2n^/10N where N is the number of points in the input image. It is important to also emphasise that when non-rigid transformation is present, the model uses all three parameters, ℛsyn,ℛreg and ℛsmooth, from Equation (13). When affine registration is present, the model focuses on using the ℛsyn and ℛreg terms (since the Laplacian operator is not activated during affine registration). 

### 3.8. Optimization

Computing Lreg requires the input of the transformed images into G, which creates a circular computing graph. Furthermore, different networks in the proposed FIRE model show different behaviours in the training process. For example, T_af_^•^^→^^•^ is more sensitive to changes in L, compared to G and F^•^^→^^•^.

To address this issue, we implemented three Adam optimisers to separately optimise parameters for (a) T_af_^•^^→^^•^, (b) T_nr_^•^^→^^•^ and (c) the synthesis encoder/two decoders. We used a uniform training procedure regardless of the size of the datasets. To optimise speed of convergence, learning rates for training T_af_^•^^→^^•^ and T_nr_^•^^→^^•^ and G/F^•^^→^^•^ were set to 10^−5^, 5 × 10^−5^ and 10^−4^, respectively. The FIRE model was trained end to end for 144,000 iterations for both datasets (empirical observation 1: for both brain and cardiac data, the total duration of the training phase was approximately 15–18 h on a Tesla P40 GPU with 24 G memory (about 7.5–9 h per dataset); empirical observation 2: for the cardiac data, model convergence was reached in the first 2–3 h, but the model was kept under training for all 144,000 iterations).

## 4. Experiments

The performance of our proposed model was evaluated using multi- (for inter-modality registration) and single- (for intra-modality registration) modal MR data.

### 4.1. MRBrainS Data

For multi-modality registration, the training data from the MRBrains13 (http://mrbrains13.isi.uu.nl/, last accessed on 1 February 2022) and the MRBrains18 (https://mrbrains18.isi.uu.nl/, last accessed on 1 February 2022) Challenges were fused. The fused dataset contained multi-modality brain MR data.

In detail, the dataset consisted of 3D T_1_-weighted, T_2_-Fluid-attenuated IR (T_2_-FLAIR) and inversion recovery (IR) data from 12 subjects, acquired using 3T MRI. The 3D T_1_, T_2_-FLAIR and IR datasets included 192, 48 and 48 slices per patient, respectively. The 3D T_1_ and IR data were already co-registered to the T_2_-FLAIR data. Images across all modalities had a voxel size of 0.958 × 0.958 × 3.000 mm^3^.

To assess model performance, manual annotations from three brain anatomical structures were used: the brain stem (BS), the cerebellum (Ce) and the white matter (WHM). For the training, validation and testing processes, we used all MRI slices from eight, one and three patients, respectively (Table 1). To perform 3D and 2D registration, all data were resampled to 1.28 mm^3^ per voxel.

In the registration process, 2D and 3D registration was performed between T_1_ and T_2_-FLAIR data, and 2D registration between IR and T_2_-FLAIR data. During training, moderate-to-strong (20–50% change across at least one dimension) affine and non-rigid transformations were randomly applied to the moving and fixed images. In the testing phase, each of the T_1_ and IR data were randomly transformed 20 times and were subsequently allowed to be registered to the corresponding T_2_-FLAIR data.

### 4.2. ACDC Data

To perform intra-modality registration, 4D cardiac cine MRI data from the 2017 ACDC (https://www.creatis.insa-lyon.fr/Challenge/acdc, last accessed on 1 February 2022) Challenge were used. Note that the fourth dimension here describes temporal resolution. The voxel size was between 1.37–1.68 × 1.37–1.68 × 5–8 mm^3^, and each 4D image has 28–40 phases covering the cardiac cycle.

In specific, the training dataset contained MRI data from 100 patients with different cardiovascular pathologies, with manual annotations of the myocardium and the left ventricle per patient at two cine-MR phases (images) per slice.

To train the FIRE model, we used all MRI phases per slice. To test the model, only the two annotated phases per slice were evaluated. For the training, validation and testing process, MRI data from 60, 10 and 30 patients were used, respectively (Table 1).

### 4.3. Evaluation Metrics and Baselines

Both the MRBrainS and ACDC datasets provided manual annotations (ground truths). To evaluate our method, we used the Dice metric to measure the overlap of the moving and fixed annotations. Increased Dice scores represent high registration performance and vice versa.

Although there are recent developments in image registration (described in the Introduction and detailed in Related Work), the mutual information (MI) and Symmetric Normalization (SyN) techniques through using the Advanced Normalization Toolbox (ANTs) (http://stnava.github.io/ANTs/, last accessed on 1 February 2022) [37] are still considered the current reference standard techniques for affine and non-rigid registration, respectively [4,9,10,11,12,13,14,15,16,17,18,30,31,32,33,34,35,36,37,38,39,40,41].

Hence, the proposed model is evaluated against the standard MI and SyN techniques to assess affine and non-rigid registration performance in multi-slice multi-modality brain and multi-phase/multi-slice single-modality cardiac MRI data, respectively.

## 5. Results

### 5.1. Inter-Modality Registration

Initially, 2D and 3D T_1_ to T_2_-FLAIR image registration was examined. The proposed model was consistent in achieving higher scores against the SyN method for all brain anatomical areas investigated (Table 2). As presented in Table 2, the proposed model outperformed the SyN method across all 2D and 3D T_1_ to T_2_-FLAIR image registration experiments examined. These results were maintained when both affine and deformable image registration were evaluated.

A visual representation of the T_1_ to T_2_-FLAIR registration is illustrated in Figure 5. In this representation, the FIRE model shows better alignment between the outer contour (in the extracerebral space) outlining the cerebrospinal fluid (shown with blue) and the actual brain tissue delineation, versus the Syn technique.

Subsequently, IR to T_2_-FLAIR registration was evaluated. Our proposed model consistently reached improved registration performance against the Syn method: a mean Dice score of 0.68 (0.3), 0.69 (0.2) and 0.70 (0.3) for the BS, Ce and WHM brain structures was reached, respectively. A visual representation of the IR to T_2_-FLAIR registration is illustrated in Figure 6.

It is important to note that the average Dice score obtained across all brain anatomical areas was below 0.45 when the SyN technique was evaluated: there was a mean Dice score of 0.43 (0.2), 0.42 (0.3) and 0.44 (0.3) for the BS, Ce and WHM brain structures, respectively. These results were repeated when a grid search for the Syn method within ANTs was carefully examined, making it impossible to derive a visual alignment of the IR to T_2_-FLAIR registration.

### 5.2. Intra-Modality Registration

Intra-modality registration was then investigated using 4D cardiac cine-MRI data. On the left ventricle anatomical areas, the FIRE model and the Syn method showed high and comparable Dice scores (Table 3, Figure 7). On the myocardial anatomical areas, the FIRE model marginally outperformed the Syn method (Table 3).

### 5.3. Computational Times

We methodologically assessed the computational costs required for each image registration process using the FIRE and the baseline SyN method implemented in ANTs. For both the FIRE model and SyN methods, the running times at the testing phase were calculated on a CPU system (Intel(R) Xeon(R) Silver 4112 CPU @ 2.60 GHz, RAM: 256 GB). Furthermore, the GPU-accelerated performance of our FIRE model was also measured using a Tesla P40 GPU with 24 G memory. Note that GPU acceleration is not available for the SyN method within ANTs.

Average running times for registering 3D and 2D data were obtained from 30 volumes, each volume containing 48 2D slices. For the 4D cardiac MR data provided in the ACDC dataset, we computed the average time for registering 10 3D volumes (of 10 slices each) at full systole to their first cardiac cine MR phase representing full diastole, therefore using 100 image pairs in total. All results are shown in Table 4.

Under the CPU mode, our FIRE model is consistently faster compared to the baseline Syn method, for both affine and non-rigid registration of the 4D cardiac data. The FIRE model shows comparable speed with the baseline in non-rigid registration of 3D and 2D brain data, whilst it was only slower for affine registration of the 3D brain data (Table 4).

When accelerated by a GPU system, our FIRE model considerably reduced the computational cost (by at least 30 times), when examined on 2D and 3D brain data. The FIRE model saved over 99.7% running time compared to the SyN method, while achieving higher accuracy in non-rigid registration of the 4D ACDC data. Bold shows improved computational time, versus the other technique. 

## 6. Discussion

We demonstrate that the FIRE method is an efficient DL model that performs accurate and fast inter- and intra-modality affine and is capable of both affine and non-rigid image registration. The FIRE model efficiently learns inverse-consistent topology-preserving deformations when evaluated on 2D and 3D multi-modal brain MR and 4D single-modal cardiac MR data, showing higher accuracy and robustness against the reference standard, the Syn method.

### 6.1. Inter-Modality Registration on Brain MRI

We showed that when both 2D and 3D image registration were examined on T_1_ to T_2_-FLAIR registration, the proposed model was consistent in showing higher performance against the Syn technique across all brain structures assessed (Table 2). This finding indicates that the proposed model can robustly model different anatomical and semantic features in these multi-tissue/multi-modal MR data; thus behaving as a brain tissue and MR modality-agnostic model (for T_1_ to T_2_-FLAIR registration).

The IR to T_2_-FLAIR image registration was compromised by the lack of anatomical information in either input modality. Implementing the Syn technique reference standard, it was not possible to reach accurate image registration results on quantitative assessments (all DICE scores were lower than 0.45), whilst we did not reach optimal alignments on visual assessments (Figure 6). It is important to note that the Syn method is the reference standard baseline technique for model evaluation in image registration [3,4,9,10,11,12,13,14,15,16,17,18,30,31,32,33,34,35,36,37,38,39,40,41]. Despite this, our FIRE model outperformed the Syn method and was able to achieve moderate mean Dice scores across all brain structures (Section 5.1).

Moreover, when examined in the testing phase, the FIRE model showed comparable speed with the baseline in non-rigid registration of 3D and 2D brain data (Table 4). The only case for which the FIRE model was slower was when affine registration of 3D brain data was measured. Nevertheless, all FIRE model-derived affine and non-rigid registration processes for all 3D brain volumes were performed in less than 40 s.

The FIRE model can therefore perform multi-modal affine and non-rigid registration on a memory-saving mode, without requiring supercomputers or GPU systems in the testing phase.

### 6.2. Intra-Modality Registration on Cardiac MRI

We demonstrated that the FIRE model can optimally model deformable transformations in cardiac cine-MR data (Table 3). Although there were only two images segmented across each cine-MR dataset (used in the testing phase), these represent the maximum geometric difference within each cardiac data-set (one segmented image is always on full diastole and the other on full systole within ACDC data). Moreover, despite the local displacements between diastole and systole being relatively small (in the 3D space), the tissue deformation is the largest within cardiac cine-MR data. In other words, this means that the cardiac tissue is non-uniformly non-rigidly deformed across phases, with the maximum deformation occurring between diastole and systole (Figure 7). Hence, while using all cardiac phases (in between diastole and systole) in the training phase (representing 4D information), our results in the testing phase show that the FIRE model can optimally learn to perform accurate registration between largely deformed geometries (different cardiac shapes between systole and diastole).

In addition, the computational cost for the FIRE model was substantially lower compared to the Syn method, when non-rigid 4D registration of cardiac MR data was examined. Thus, the FIRE model demonstrated overall improved registration speed when dealing with 4D data, which can be particularly important across numerous cardiac MR applications involving time series [41,42,43,44]. To our knowledge, the only other technique that can effectively perform inverse-consistent registration directly in the training phase is the CycleMorph model [33]. In our work, we define the output deformation fields with simple inverse-consistency constraints, devising a model that can inherently learn topology-preserving image registration in a consistent mode (via the synthesis encoder G and a single transformation network).

### 6.3. FIRE Model Explainability

Despite recent developments in DL, inter-modality image registration is not yet widely investigated using deep networks [30,31,33,34]. A major reason is the absence of explainability regarding why a DL model learns and/or where it fails to learn anatomical image information and semantics. This becomes more evident in image registration, as the lack of standardised annotations (ground truths) may have diversified methods for model evaluation and thus, discouraged model explainability [5,6,7,26,27,28,29,30,31,32,33,34,35,38,41]. Our main consideration here is that DL explainability can help to interpret model-data interrelationships before (or beyond) deriving class activation maps [45,46], which can encourage thorough (and explainable) DL investigations in inter-modality image registration. We showed that the FIRE model can learn anatomical and semantic representations across modalities, and demonstrated improved performance for inter-modality image registration, versus the SyN method.

It is important to note that T_1_, T_2_-FLAIR and IR are routinely used MR sequences, but there are fundamental disparities in terms of the MR physics involved and, thus, their imaging content [47]. These MR modalities are optimised to provide different imaging information in clinical MRI: T_1_ is designed to provide detailed anatomical information, whereas IR and T_2_-FLAIR are primarily implemented to extract functional information [47]. In T_1_ to T_2_-FLAIR registration, our cross-domain synthesis component (encoder/decoders) can effectively leverage anatomical information from the T_1_ data and can subsequently map this information to complementary functional T_2_-FLAIR information. This complementary anatomical-functional information is compromised in the IR to T_2_-FLAIR data registration during cross-domain synthesis. Hence, not the anatomical information per se, but the presence of (heterogeneous) complementary anatomical-functional information makes the multi-modal transformations invertible in relation to each other, which in turn encourages our model to produce inverse-consistent topology-preserving image registrations. Although not in the scope of the current study, further work is underway to quantify this complementary information in our model, through gradient-weighted class activation mapping (Grad-CAM) [48], across each of the model main components (enconder, STN, decoders).

We should also highlight that there can be multiple methodologies by which it can become possible to enrich “poor” anatomical information within a dataset, towards enhancing inverse-consistency through complementary information. For example, an accurate semi-supervised inter-modality learning model could be explored to derive anatomical annotations and guide the registration process [49,50], hence, potentially adapting our technique for functional (non-anatomical) MR modalities [49,50]. Other active contour-based segmentation algorithms for fast image segmentation have shown excellent segmentation accuracies even for images with large noise interference and intensity inhomogeneities [51,52,53], and therefore may be applicable to enrich anatomical information in multi-modal registration of imaging modalities with decreased anatomical information. 

## 7. Conclusions

We have demonstrated a robust method that efficiently models inter- and intra-modality image registration through bi-directional cross-domain synthesis and factorised spatial transformation. To our knowledge, parallelising cross-domain synthesis while modelling spatial transformation to dynamically learn anatomical and latent representations simultaneously across modalities has not been previously investigated.

Our work showed the efficiency of our methodology as we improve intra- and inter-modality registration by maintaining fast computational times in the testing phase, across all registration tasks.

In broader terms, the main significance of our work is the simultaneous bi-directional cross-domain image synthesis and spatial transformation per synthesis channel, which enables us to diversify our technique across numerous multi-modality medical image analysis scenarios. Through providing a model explainability framework, we can suggest that it is possible to adapt our technique to learn anatomical-semantic information when we have at least one dataset with enriched anatomical information. This means that we can customise our methodology to address image registration on additional anatomically enriched imaging data, such as with computed tomography (CT) or ultrasound data, next to MRI.

To date, the main limitation of our work is that we have only assessed multi-modal MRI data. However, we examined two organ areas (brain and heart), intra- and inter-modality registration, and investigated 2D, 3D and 4D registrations frames. This inspired our future work as we are currently investigating model generalisation frameworks to allow further multi-organ and multi-modal applications. In conclusion, we show here that our methodology demonstrated improved performance and efficiency against the current standard Syn method, thus presenting a versatile image registration technique that may have merit in the clinical setting.

## Figures and Tables

**Figure 1 sensors-22-02125-f001:**
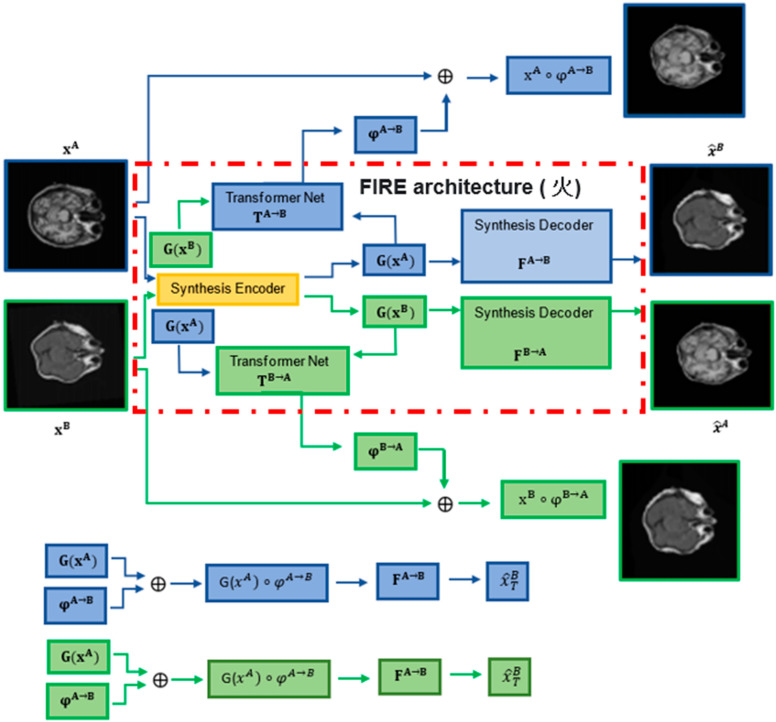
FIRE model (**火**, **Chinese symbol for “fire”**) architecture (red dash-dotted line box). Two synthesis encoders G(x^A^) and G(x^B^) extract modality-invariant (latent) representations. Two synthesis decoders F^A→B^ and F^B→A^ map the representations extracted by G(x^A^) and G(x^B^) to $\hat{x}$^B^ and $\hat{x}$^A^ (images synthesised), respectively (see F^A^^→B^ and F^B→A^ at the bottom). Finally, two transformation networks T^A^^→B^ and T^B→A^ model the transformation fields, finalising the FIRE training process.

**Figure 2 sensors-22-02125-f002:**
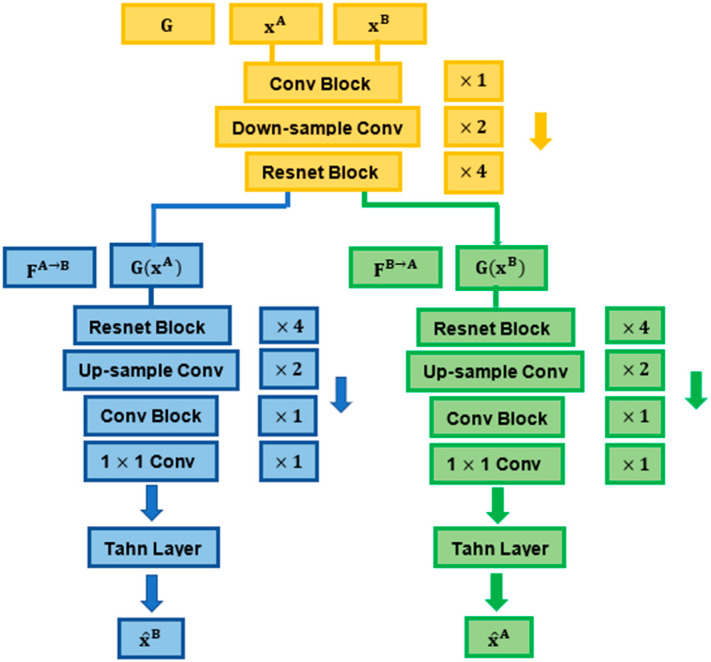
Details of the encoder and decoders used for the synthesis process.

**Figure 3 sensors-22-02125-f003:**
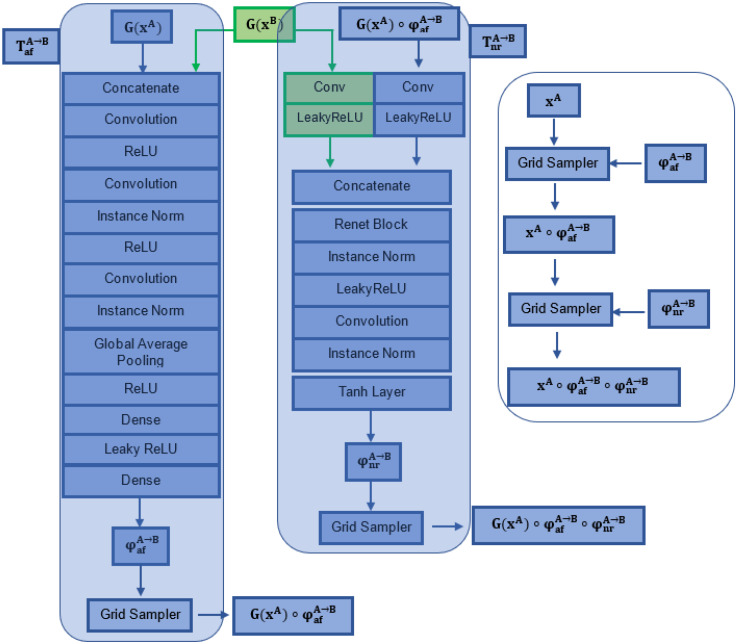
Affine and non-rigid spatial transformation networks incorporated in the FIRE model.

**Figure 4 sensors-22-02125-f004:**
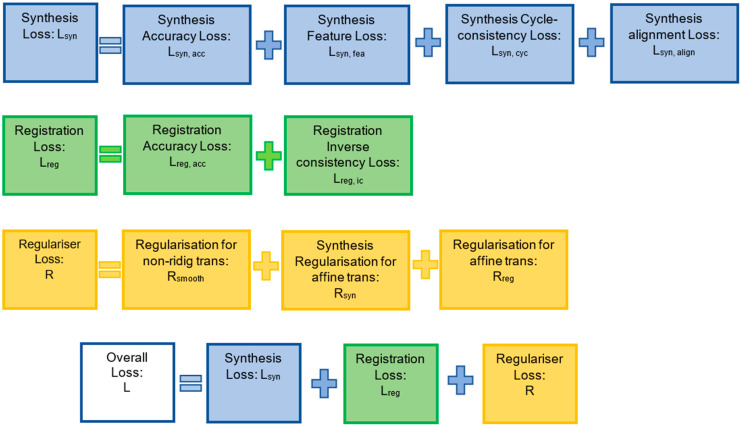
Graphical description of all losses defined in the FIRE model. Full descriptions and mathematical details are provided in Section 3.5, Section 3.6 and Section 3.7 (in Methods).

**Figure 5 sensors-22-02125-f005:**
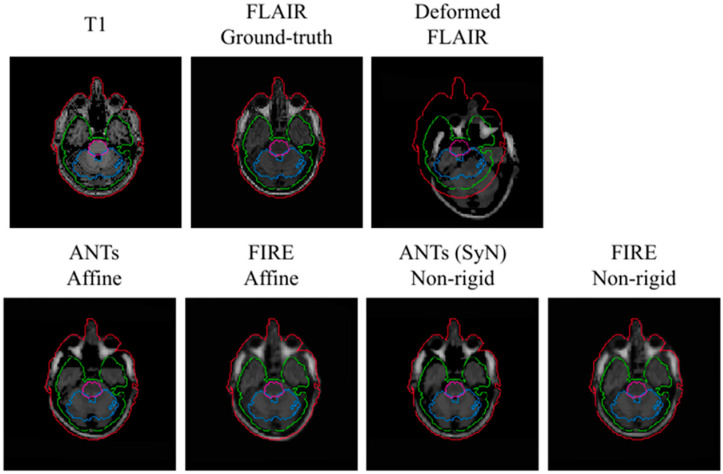
Visual representation of the T_1_ to T_2_-FLAIR registration using the MRBrainS data. Brain stem, cerebellum and white matter used to assess registration performance are illustrated with purple, blue and green, respectively. Red illustrates the outer contour outlining the entire head.

**Figure 6 sensors-22-02125-f006:**
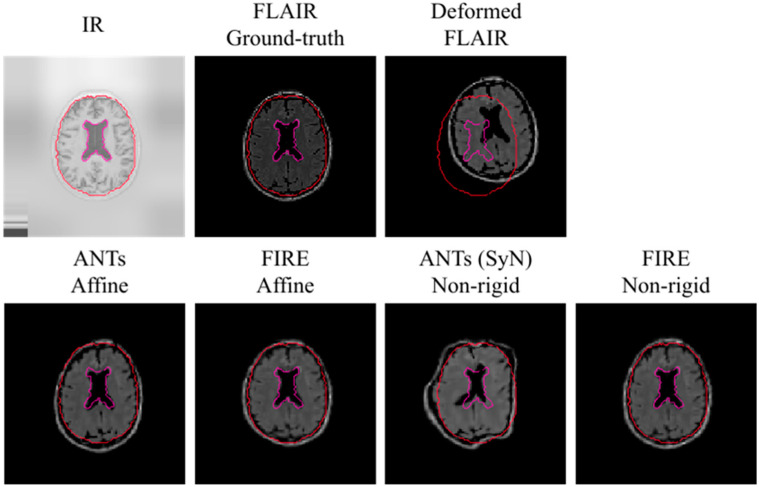
Visual representation of the IR to the T_2_-FLAIR registration using the MRBrainS data. To increase clarity because of having less anatomical information in IR and T_2_-FLAIR data, we use the outer brain region and ventricles (with red and purple respectively) to visually demonstrate registration results. Note that the upper left image shows an IR image before pixel normalisation occurred during model fitting.

**Figure 7 sensors-22-02125-f007:**
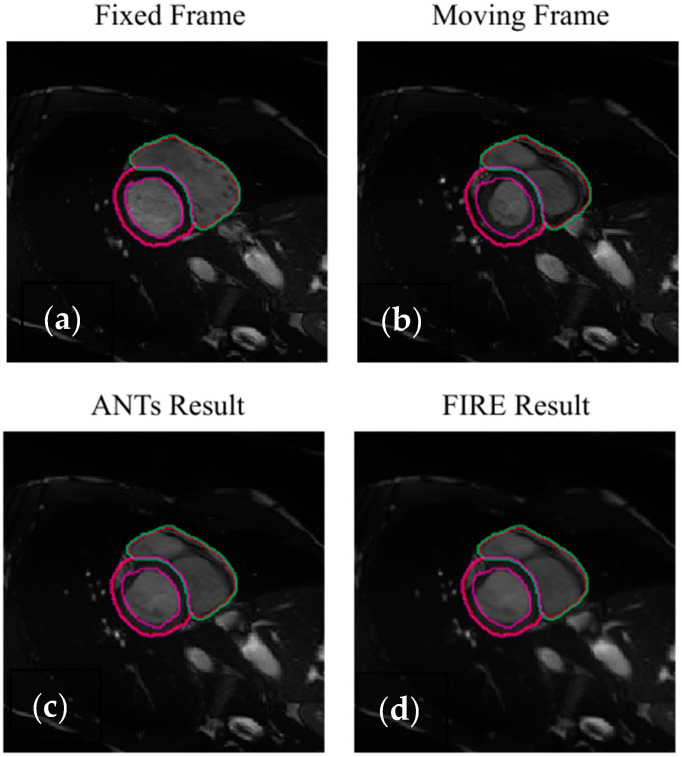
Illustration of the intra-modality image registration using ACDC data (the purple and magenda contours demonstrate the endocardial and epicardial left ventricle delineations, respectively). The red and green contours illustrate the right ventricle delineations, respectively (not used in the analysis due to showing small changes across the cardiac cycle). Full diastole (**a**,**c**) and systole (**b**,**d**) are illustrated.

**Table 1 sensors-22-02125-t001:** Total number of images for training, validation and testing per brain and cardiac dataset. Note that for the cardiac data, only two images per cardiac slice contained annotations and were evaluated at testing.

Organ	Sequence	Training	Validation	Testing
Brain	T_1_	1536	192	576
	IR	384	48	144
	T_2_-FLAIR	384	48	144
Cardiac	Cine	14,400	240	480

**Table 2 sensors-22-02125-t002:** Results obtained from the 2D and 3D T_1_ to T_2_-FLAIR registration, on the MRBrainS data. Dice scores were measured on the brain stem (BS), cerebellum (CE) and white matter (WHM). Standard deviations are included within the parenthesis. Bold shows our FIRE model-derived results.

Data	Object	Unaligned	ANTs-Affine	FIRE-Affine	ANTs-SyN	FIRE
2D	BS	11.62 (6.1)	61.25 (3.7)	62.90 (4.1)	78.73 (7.3)	**80.68 (7.7)**
	CE	7.17 (4.4)	63.32 (3.2)	64.36 (4.0)	75.72 (8.1)	**76.96 (7.3)**
	WHM	14.29 (7.5)	59.12 (4.5)	59.97 (4.4)	81.36 (6.0)	**84.18 (3.7)**
3D	BS	27.15 (9.2)	67.15 (3.1)	69.81 (4.1)	79.77 (6.7)	**81.08 (7.0)**
	CE	28.38 (9.5)	68.38 (3.6)	70.62 (3.7)	86.00 (6.9)	**86.13 (7.2)**
	WHM	20.27 (9.3)	60.27 (3.8)	60.61 (3.8)	72.33 (7.4)	**72.56 (7.1)**

**Table 3 sensors-22-02125-t003:** Results on ACDC data. Dice scores computed on left ventricular endocardium (LVe) and myocardium (Myo). Standard deviations are included within the parenthesis. LVe: left ventricle, Myo: myocardial tissue. Bold shows higher model performance, versus the other model.

Object	Unaligned	ANTs-SyN	FIRE
LVe	65.75 (16.2)	**90.81 (4.3)**	90.28 (5.5)
Myo	51.95 (14.5)	70.71 (5.6)	**71.86 (6.3)**

**Table 4 sensors-22-02125-t004:** Mean computational times of the FIRE and the Syn method in the testing phase. The parentheses show standard deviations.

Data	Registration	Multi-Modal	Number of Images Used	FIRE-GPU (Sec)	FIRE-CPU (Sec)	SyN (Sec)
MRBrainS (2D)	Affine	Yes	1440	0.305 (0.01)	1.1856 (0.03)	**0.3375 (0.06)**
	Non-rigid	Yes		0.352 (0.01)	1.5370 (0.04)	**1.3987 (0.09)**
MRBrainS (3D)	Affine	Yes		0.454 (0.06)	37.939 (0.48)	**4.3542 (0.13)**
	Non-rigid	Yes		0.555 (0.07)	**29.004 (0.34)**	30.069 (0.49)
ACDC (4D)	Affine	No	200	4.873 (0.08)	**199.54 (1.26)**	224.78 (1.89)
	Non-rigid	No		5.026 (0.09)	**200.16 (1.78)**	1887.30 (2.25)

## Data Availability

For our data analysis, we used publicly available datasets and we explicitly describe how to access these data. Code will be made available at GitHub.

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
