# Peer review of "Unsupervised Image Registration towards Enhancing Performance and Explainability in Cardiac and Brain Image Analysis"

_sensors, 2022, doi:10.3390/s22062125_

Round 1

Reviewer 1 Report

I really enjoyed reading the manuscript by Wang et al. on the use of DL networks for both affine and non-rigid registration. The authors make a very strong point of introducing a flexible and computationally efficient registration tool. 

While I think the that the work is very solid and worthy of publication, I have some minor concerns about it. I really liked their overview of the background, but found their description of their actual technique rushed. I would have loved to see some more detail on the presented architecture (e.g. actual size of layers). I am under the impression that the manuscript is a bit math heavier than average for the readership of Sensors, so I would encourage some more explanation for the choice of the loss and regularization functions to make it more accessible. 
With respect to the results, it would help if Fig. 4 and 5 had more detailed captions, directing the reader to what exactly we are looking at, especially for the outlines. It seems like the authors have run a variety of tests for which I could not find an explanation in the text. Also, the IR MRI in figure 5 looks strange. In table 2, the Dice coefficients in the 70s range need to be discussed to be put into context with respect to state of the art. 

Lastly, I was very intrigued by the "explainability" in the title of the article. However, reading the associated section 6.3., I fail to understand the point the authors want to make; the only thing I am reading is that the higher anatomical detail in T1 weighted MRI provides a better reference for registration, which should be common knowledge to anybody that has done brain MRI registration. 

Overall, I think the paper presents a very interesting work worthy of publication, but some more effort needs to be taken into making its presentation both more robust and accessible. 

Reviewer 2 Report

The authors developed and tested an unsupervised Deep Learning (DL) technique to learn image representation and spatial transformations through image synthesis, possibly useful for numerous medical images scenarios. The unsupervised learning methods do not need manual annotations for model training, saving work and time. The developed method (named "FIRE") had better performance than Advanced Normalization Toolbox (ANTs), considered the current standard technique on multi-modality image registration.

The authors modified the CycleGAN generator architecture by splitting it in the middle (encoder and decoder), but they did not explain why they made these changes.

The authors must explain each process of the Figure 2 flowchart, for example, downsampling convolutional layer, describe the purpose of this process.

Each process of Figure 3 needs a description.

A flowchart specific for Loss functions would be more interesting to understand the training procedure.

It seems that subsections 3.5, 3.6, 3.7, and 3.9 are subsections of 3.4 Training Procedure and Loss functions.

In Figure 6, the authors must identify which image is diastole a) and systole b).

The authors must present a table with the total number of images used for training the model and used to test the model.

The authors could show the time used for the training phase of the DL and how many iterations were needed.

The authors need to show how many images were processed to show the results of Tables 1 and 2.

The authors should put a column with the number of slices used to calculate the mean computational times presented in Table 3.

Reviewer 3 Report

This paper employs unsupervised image registration to enhance the performance and explainability in cardiac and brain image analysis. In general, this paper needs to follow the comments below to improve the quality:

  1.  The model used in the article should be described in more detail.
  2. The text introduction format and layout of the graphics are inconsistent.
  3. The formula structure layout of the article is inconsistent.
  4. The understanding of research directions in abstracts and keywords should be appropriately expanded.
  5. When introducing relevant work, the contents mentioned in abstracts should be introduced more.
  6. Some schematic diagrams can be added to explain the introduction of the conclusion.
  7. Some text in the article table is not aligned.
  8. Recent developed image processing methods should also be reviewed in the introduction section such as:  Active contours driven by region-scalable fitting and optimized laplacian of gaussian energy for image segmentation, Signal Processing; Active contours driven by adaptive functions and fuzzy c-means energy for fast image segmentation, Signal Processing; A level set method based on additive bias correction for image segmentation, Expert Systems with Applications

Round 2

Reviewer 3 Report

The authors have addressed all my comments.